# The Covert Side of Ascites in Cirrhosis: Cellular and Molecular Aspects

**DOI:** 10.3390/biomedicines13030680

**Published:** 2025-03-10

**Authors:** Carlo Airola, Simone Varca, Angelo Del Gaudio, Fabrizio Pizzolante

**Affiliations:** 1CEMAD Centro Malattie dell’Apparato Digerente, Fondazione Policlinico Universitario Agostino Gemelli IRCCS, 00168 Rome, Italy; simone.varca01@icatt.it (S.V.); angelo.delgaudio01@icatt.it (A.D.G.); 2Facoltà di Medicina e Chirurgia, Università Cattolica Sacro Cuore, Largo Agostino Gemelli, 8, 00168 Rome, Italy

**Keywords:** ascites physiopathology, lymphangiogenesis, macrophages, inflammation, paracrine signaling

## Abstract

Ascites, a common complication of portal hypertension in cirrhosis, is characterized by the accumulation of fluid within the peritoneal cavity. While traditional theories focus on hemodynamic alterations and renin–angiotensin–aldosterone system (RAAS) activation, recent research highlights the intricate interplay of molecular and cellular mechanisms. Inflammation, mediated by cytokines (interleukin-1, interleukin-4, interleukin-6, tumor necrosis factor-α), chemokines (chemokine ligand 21, C-X-C motif chemokine ligand 12), and reactive oxygen species (ROS), plays a pivotal role. Besides pro-inflammatory cytokines, hepatic stellate cells (HSCs), sinusoidal endothelial cells (SECs), and smooth muscle cells (SMCs) contribute to the process through their activation and altered functions. Once activated, these cell types can worsen ascites accumulationthrough extracellular matrix (ECM) deposition and paracrine signals. Besides this, macrophages, both resident and infiltrating, through their plasticity, participate in this complex crosstalk by promoting inflammation and dysregulating lymphatic system reabsorption. Indeed, the lymphatic system and lymphangiogenesis, essential for fluid reabsorption, is dysregulated in cirrhosis, exacerbating ascites. The gut microbiota and intestinal barrier alterations which occur in cirrhosis and portal hypertension also play a role by inducing inflammation, creating a vicious circle which worsens portal hypertension and fluid accumulation. This review aims to gather these aspects of ascites pathophysiology which are usually less considered and to date have not been addressed using specific therapy. Nonetheless, it emphasizes the need for further research to understand the complex interactions among these mechanisms, ultimately leading to targeted interventions in specific molecular pathways, aiming towards the development of new therapeutic strategies.

## 1. Introduction

Ascites is a pathological condition characterized by the accumulation of fluid within the peritoneal cavity. Commonly found as a complication of portal hypertension in cirrhosis, ascites represents the main decompensation event in this setting [1]. Ascites development is associated with the dysregulation of systemic homeostasis and is frequently followed by refractory ascites, hepato-renal syndrome, or hepatic encephalopathy, which is correlated with an increased mortality in these patients despite the appropriate therapy [1,2]. Understanding the intricate molecular mechanisms that contribute to the development of ascites is essential for improving diagnosis, treatment, and patient outcomes. The pathophysiological mechanisms underlying ascites formation in patients affected by cirrhosis include changes in the oncotic and hydrostatic pressures in the systemic and splanchnic circulation [3,4]. Equally, the renin–angiotensin–aldosterone system’s (RAAS) activation raises systemic vascular resistance and encourages sodium and water reabsorption in the kidneys, causing fluid retention and aggravating ascites development [5]. Furthermore, the decrease in effective arterial blood volume observed in conditions such as liver cirrhosis activates the sympathetic nervous system, which triggers the release of antidiuretic hormone (ADH) from the posterior pituitary gland, enhancing water reabsorption in the collecting ducts of the kidneys, further contributing to fluid retention [6]. Ascites development is also significantly influenced by oxidative stress and inflammation. As shown in liver cirrhosis, chronic inflammation activates inflammatory cells including neutrophils and macrophages, which release pro-inflammatory cytokines and reactive oxygen species (ROS) [7]. These inflammatory mediators enhance arterial permeability and endothelial dysfunction, which promote fluid and protein extravasation into the peritoneal cavity. ROS can harm the endothelial cells that line the blood arteries directly, aggravating the vasculature’s leakiness [7]. Given that the pathophysiological mechanisms underlying ascites formation have been studied for more than 100 years [8], recently, significant progress has been made in elucidating the complex network of molecular events involved in ascites formation, highlighting the contribution of other factors, such as cytokines, growth factors, hepatic and immune cells, and lymphangiogenesis. This review aims to provide a comprehensive overview of these mechanisms, focusing on the usually less considered elements which could be involved in ascites formation (Figure 1).

## 2. Nitric Oxide (NO) and Vascular Endothelial Growth Factor (VEGF)

NO is a gaseous signaling molecule produced by endothelial cells, macrophages, and hepatocytes and exerts diverse physiological effects and plays a crucial role in maintaining vascular homeostasis. In the context of ascites, NO has both pro- and anti-inflammatory properties, making it a double-edged sword [9]. NO is known to induce vasodilation and increase vascular permeability. These effects are mediated through the activation of soluble guanylate cyclase (sGC) and the subsequent elevation of intracellular cyclic guanosine monophosphate (cGMP) levels. Increased cGMP leads to the relaxation of the vascular smooth muscle cells and widening of the blood vessels, resulting in increased capillary permeability. Therefore, it contributes to the leakage of plasma proteins and the subsequent accumulation of fluid in the peritoneal cavity [10]. Furthermore, NO activity seems to be closely related to angiogenesis. In fact, angiogenesis plays a critical role in sustaining the accumulation of ascitic fluid in the peritoneal cavity. The NO-mediated upregulation of VEGF expression enhances the formation of new blood vessels, thereby facilitating the influx of fluid and solutes into the peritoneal cavity [11]. Conversely, in 2006, Abraldes et al. examined the endothelial NO synthase (eNOS), VEGF, and systemic and splanchnic hemodynamics in rats with various degrees of portal hypertension. In this preclinical study, it was shown that even moderate degrees of portal hypertension increased vascular endothelial growth factor (VEGF) synthesis and consequently the eNOS expression in the intestinal microcirculation [12]. In another preclinical study, gene transfer techniques were performed in a cirrhotic rat population to selectively inhibit the eNOS activity in the systemic circulation. As a result, a reduction in the volume of ascites was obtained, together with an improvement in renal excretion function [13]. Moreover, other cell types classically not included among those involved in ascites formation, such as lymphatic endothelial cells (LECs), produce NO, with an effect on peritoneal fluid accumulation [14]. Indeed, LECs inhibit smooth muscle cell (SMC) activity and recruitment by producing NO, reducing their contractility and consequently the fluid reabsorption through the lymphatic system [14]. On the contrary, NO interacts with the RAAS by inhibiting the production of aldosterone and promoting natriuresis and diuresis and the potential alleviation of ascites [15]. Thus, NO seems to participate in several mechanisms involved in ascites development, as an enhancing factor but also as a protective one. Nonetheless, this ambivalent role concerning portal hypertension and ascites formation suggests a dependance on other mechanisms and interaction with other molecules and cells. The main mechanisms mentioned are summarized in Table 1.

## 3. Cytokines, Signaling Molecules That Orchestrate Cells in Ascites Development

In ascites pathogenesis, other agents and signaling pathways also contribute significantly. Cytokines induce endothelial cell activation and disrupt the endothelial barrier, further increasing vascular permeability and contributing to ascitic fluid accumulation [16]. Additionally, some cytokines, such as interleukin 6 (IL-6) and interferon γ (IFN-γ), directly activate the RAAS [17]. Indeed, pro-inflammatory cytokines, such as tumor necrosis factor-alpha (TNF-α) and IL-6, are elevated in ascitic fluid. Nonetheless, higher levels of TNF-α in the plasma and ascitic fluid of patients with spontaneous bacterial peritonitis (SBP) correlate with a higher hepatic venous pressure gradient (HVPG) [18]. Cytokines are particularly involved in the formation of inflammatory ascites through local inflammatory action, and they have mostly been studied in cancer-related ascites [19,20]. Indeed, the IL-4, IL-6, and IL-10 levels were found to be higher in patients with ascites and hepatocellular carcinoma (HCC) than patients with cirrhosis alone [21]. Nonetheless, systemic inflammation is one of the principal cirrhosis decompensation events. In fact, cytokine cascade has been demonstrated to be involved in non-neoplastic ascites development as well [2]. The IL-1α, IL-6, and TNF-α levels were found to be higher in the plasma and ascites of cirrhotic patients compared to those in plasma from healthy individuals [22]. Another clinical observation highlighted the local role of cytokines in ascites formation in patients with alcohol-related cirrhosis. An increase in ascitic IL-6 and IL-12 levels was found to promote ERK1/2 phosphorylation in the peritoneal macrophages, with no effect on the peripheral blood [23]. In addition to this, cytokines participate as signaling molecules in several cell interactions; among these, hepatic stellate cells (HSCs), sinusoidal endothelial cells (SECs), and SMCs are particularly involved in ascites development [24]. The main mechanisms mentioned are summarized in Table 2.

## 4. Hepatic Stellate Cell (HSC) and Sinusoidal Endothelial Cell (SEC) Crosstalk

The interaction between SECs and HSCs plays a crucial role in ascites formation [25]. HSCs, also known as perisinusoidal cells or Ito cells, are located in the space of Disse within the liver sinusoids [26]. Under normal physiological conditions, the HSCs remain quiescent and store vitamin A [26]. However, in response to liver injury or chronic inflammation, HSCs undergo activation, transitioning into a myofibroblast-like phenotype [27]. Activated HSCs are the major source of extracellular matrix (ECM) proteins such as collagen, fibronectin, and laminin which contribute to the fibrotic changes observed in liver diseases [28,29]. Concerning portal hypertension, higher blood levels of increased ECM deposition markers such as type I-III-IV-V-VI collagen, biglycan, and elastin are associated with a higher HVPG [30]. Activated HSCs interact with many cell types through paracrine signals. The crosstalk between HSCs and SECs is bidirectional and involves the secretion of various soluble mediators, including cytokines, growth factors, and vasoactive substances, contributing to the formation of ascites through multiple mechanisms [31]. Firstly, activated HSCs secrete profibrogenic factors, such as transforming growth factor-beta (TGF-β), platelet-derived growth factor (PDGF), and connective tissue growth factor (CTGF). These molecules stimulate the SECs to produce endothelin-1 (ET-1) and NO. ET-1 is a potent vasoconstrictor and leads to increased intrahepatic resistance by causing vasoconstriction of the sinusoids. On the other hand, NO has vasodilatory effects and can counterbalance the vasoconstrictive actions of ET-1, as previously described [31]. Moreover, the HSC and SEC interaction influences the deposition of ECM proteins within the liver. Activated HSCs secrete matrix metalloproteinases (MMPs) that degrade the ECM, allowing for the remodeling and deposition of new matrix components. SECs, in turn, secrete tissue inhibitors of metalloproteinases (TIMPs) to regulate the activity of the MMPs. In conditions of liver disease, the balance between MMPs and TIMPs is dysregulated, resulting in excessive ECM deposition, and contributes to the development of liver fibrosis and portal hypertension, which further exacerbates ascites formation [32]. The altered secretion patterns of activated SECs affect the vascular tone and integrity within the liver. In addition to ET-1 and NO, SECs produce various other vasoactive substances, including prostaglandins, endothelial-derived hyperpolarizing factor (EDHF), and angiotensin II. These factors modulate the contractility of the HSCs and sinusoidal tone, ultimately impacting intrahepatic resistance and the development of ascites [33]. The TGF-β/Smad signaling pathway seems to have a crucial role in this interaction. The TGF-β released by activated HSCs stimulates the expression of profibrogenic genes in SECs, leading to the increased production of ECM proteins [34]. Additionally, TGF-β promotes the differentiation of the SECs into a pro-inflammatory and pro-fibrotic phenotype, further contributing to the pathogenesis of ascites [35]. Nonetheless, TGB-β, which is upregulated in cirrhosis, is associated with hepatic microcirculation disruption, worsening portal hypertension [36]. Another important factor in the crosstalk between HSCs and SECs is the presence of reactive oxygen species (ROS). ROS are generated during liver injury and activate signaling pathways in both cell types [37]. ROS induce the release of pro-inflammatory cytokines and growth factors from HSCs, promoting the activation and proliferation of SECs. In turn, activated SECs produce ROS, creating a feedback loop that perpetuates the inflammatory and fibrotic processes [38]. ROS also increase vascular permeability by inducing the phosphorylation of PYK2 (p-PYK2) and VE-cadherin (p–VE-cad) in the SECs and reducing cell adhesion, exacerbating inflammatory ascites formation [39]. Moreover, recent studies have highlighted the role of microRNAs (miRNAs), which have been identified as critical regulators of HSC activation and HSC-SEC crosstalk mediators [38]. Indeed, the miRNAs released by activated HSCs can be taken up by SECs, altering their gene expression and promoting their pro-fibrotic phenotype [38]. The main mechanisms mentioned are summarized in Table 3.

## 5. Macrophages: Neglected Cells in Ascites Formation?

In cirrhosis, macrophage function, as well as that of the other immune cells, is profoundly disrupted, leading to both hyperinflammation and a condition similar to the immune paralysis observed in sepsis [40,41]. These aspects can be present at the same time in the same individual and even in the same organ, representing two extremes of the spectrum that characterizes cirrhosis responsible for the great complexity of the course of the disease [42,43]. Macrophages are involved in several of the above citated pathways related to ascites. Indeed, a subset of the tissue-resident GATA6^+^ macrophage population is physiologically present in peritoneal cavity fluids, the so-called peritoneal macrophages (PMs), being the most represented cell population in the peritoneal fluids [23,44]. Under physiological conditions, PMs continually pass through the peritoneum–blood barrier and phagocytize cellular and molecular fragments [45]. PMs have recently been divided into two subclasses, large PMs (CD206^+^CD163^+^) and small PMs (CD206^−^) [46]. Large PMs are the predominant subset, as they express anti-inflammatory genes more frequently, while small PMs are the minority, becoming the prevalent subpopulation in response to inflammation [47]. It has been shown that under inflammatory conditions, large PMs disappear from the peritoneum [48]. It is possible that they do not just migrate but rather differentiate into other cell types. In fact, inflammation induces the transdifferentiation of CD11b^+^ PMs into LECs, enhancing abnormal lymphangiogenesis and altering lymphatic drainage [49,50]. PM transdifferentiation is associated with a higher expression of lymphangiogenic genes, such as VEGF-A164, VEGF-A120, VEGF-C, VEGF-D, and Ang-2, and pro-inflammatory cytokines, such as TNF-α, IL-1β, and IL-6 [50]. Usually, PMs increase the IL-6 production following inflammatory exogenous stimuli such as LPS exposition [51]; however, higher levels of IL-6 have been found in ascites patients with cirrhosis and no previous spontaneous bacterial peritonitis [52]. Additionally, the increased production of IL-6 has been associated with higher polarization of macrophages to the CD206^+^ phenotype [53]. Nonetheless, the large PMs derived from the circulating monocytes migrating to the peritoneum after inflammation display an anti-inflammatory phenotype, with high production of IL-10 [54]. It is crucial to note that in patients with cirrhosis, the balance between macrophage inflammatory and an anti-inflammatory phenotype is constantly menaced. Indeed, portal hypertension causes a significant disbalance between the intestinal barrier and the gut microbiota that leads to the continuous passage of pathogen-associated molecular patterns (PAMPs) in the bloodstream to the liver and to the peritoneum [55]. Thus, the immune cells in the liver and peritoneum are overstimulated and activated, even in the absence of a clinical inflammatory manifestation. Concerning PMs, their phenotype could also change with regard to the etiology of liver disease, as demonstrated by a clinical trial evaluating this specific macrophage subpopulation in patients with alcohol- and HCV-related cirrhosis. In an alcohol-related group, PMs showed a marked pro-inflammatory profile, while in HCV-related cirrhosis, pro-inflammatory cytokines were less expressed in the PMs. Nonetheless, if exposed to bacterial antigens, even in HCV-related cirrhosis, their pro-inflammatory cytokine production was significantly increased through the extracellular signal-related kinase 1 and 2 (ERK1/2) pathway [56]. On the contrary, a preclinical analysis showed that the PMs in rats with metabolic dysfunction, such as obesity and type 2 diabetes, showed increased production of citrulline and NO, with an anti-inflammatory effect, and had a reduced capacity for cytokine secretion [57,58]. However, in the complexity of the cellular interplay between the peritoneum, the portal system, and the liver, the role of PMs cannot be isolated. In fact, PMs seem to also have a strong relationship with other hepatic macrophages, such as tissue-resident, monocyte-derived macrophages, and Kupffer cells. Preclinical studies have shown that in cases of liver damage, GATA6^+^ PMs directly migrate to the liver through the mesothelium and the liver capsule, interacting with hepatic macrophages [59,60,61]. Equally, participating in the inflammatory response within the liver tissue, PMs and hepatic macrophages produce an amount of pro-inflammatory cytokines that may be involved in ascites formation [62]. Nonetheless, the hepatic macrophages also produce non-inflammatory factors which act in endocrine signaling. Indeed, it has been shown that in obese mice, the liver macrophages produce insulin-like growth factor 7 (IGFBP7), which is involved in the systemic action of metabolism modulation [63]. Interestingly, IGFBP7 has been identified as a potential inducer of lymphangiogenesis [64]. While the involvement of the PMs and hepatic macrophages in non-inflammatory ascites remains unclear, their potential role as key players is intriguing (Figure 2). Their remarkable plasticity, migratory ability, and diverse molecular production make them ideal candidates for orchestrating the complex cellular and molecular interactions that lead to fluid accumulation in the peritoneal cavity. Several mechanisms could be at play, such as NO and cytokine production and lymphangiogenesis modulation. Nonetheless, other mononucleated immune cells, such as NK cells, could be involved in ascites. Indeed, their role in preventing the translocation and the spread of intestinal bacteria in ascites has been demonstrated [65]. However, specific genes could be involved, as shown by Legaz et al., who found that the absence of the KIR2DL2 and KIR3DL1 genes, usually expressed by the NK cells, predisposed patients with an alcohol use disorder to the development of ascites [66]. Further research is needed to explore this hypothesis and understand the specific role of each of these mechanisms according to which the PMs and hepatic macrophages might contribute to non-inflammatory ascites formation. The main mechanisms mentioned are summarized in Table 4.

## 6. The Lymphatic System and Lymphangiogenesis: Drainage Impairment in Ascites Pathogenesis

The lymphatic system plays a crucial role in maintaining fluid balance, immune function, and lipid absorption. Excess fluid and proteins are removed from the abdominal and peritoneal cavities via the lymphatic vessels. A dysfunction in the lymphatic system can lead to the impaired drainage of fluid from the tissues, including the abdominal cavity, contributing to the formation of ascites. The increased intrahepatic pressure in liver cirrhosis can also cause the liver’s lymphatic capillaries to become compressed, further impairing lymphatic outflow. As a result, the buildup of fluid in the peritoneal cavity is exacerbated, which helps to cause ascites [67]. The lymphatic system is characterized by blind-ended tubular structures with a unidirectional flow, lacking a central pump, as is the case in the cardiovascular system. In the peritoneum, below a single layer of mesothelial cells, there is an extremely thin and discontinuous connective tissue layer and a layer of fenestrated lymphatic vessels [68].

This system has several roles: regulating fluid homeostasis, participating in the nutrient absorption in the gastro-intestinal tract, and immune surveillance [69]. This system is also involved in pathogen removal and the local defense against bacterial invasion through the activation of resident cells and the recruitment of circulating immune cells [70,71]. The lymphatic capillaries consist of a single layer of LECs. SMCs and pericytes are found only in the collection of lymphatic vessels [72]. Lymphatic vessels are anchored to the surrounding connective tissue by very thin fibers [73]. When fluids accumulate in the interstitial tissue, the lymphatic vessels fill with lymph and SMCs and, activated by mechanical stretching, contribute to the increased propulsive force in the lymph/lymphatic circulation. Furthermore, Ribera J et al. suggest that the LECs also produce NO, thus inhibiting the SMCs, as happens in the blood vessels, whose contractility is reduced by NO, as is SMC recruitment, creating negative feedback [14]. Besides the peritoneal lymphatic system, there is also a hepatic lymphatic system that may be involved in ascites formation. Yamauchi et al. documented how the enlargement and increased density of the lymphatic vessels in the liver correlate with the severity of fibrosis/cirrhosis, with which increased production of VEGF C and D in the fibrotic liver tissue is associated [74]. This has been documented both in humans and in a CCl4-induced rat model of cirrhosis. In Yamauchiy et al., the authors demonstrated that the intrahepatic lymphatic vessels are abundant and enlarged in cirrhotic patients [74]; this phenomenon is thought to be due to increased lymph production, which is caused by the dysregulation of the drainage of vascular flow from the sinusoid to the central or terminal hepatic veins in cirrhosis [74]. No differences were observed in hepatitis but were found in fibrosis and cirrhosis. So, the lymphatic vessels in the liver increase in size and number with the progression of chronic fibrosis [74]. Although these changes initially act as a compensatory mechanism, with the progression of liver disease, the increase in the lymph flow overpasses the capacity of reabsorption, with an increase in lymphatic pressure inducing mechanical stress in the lymphatic vessels and causing lymphangiectasia [75]. This can lead to the rupture of the lymphatic vessel and to the consequent loss of protein, chylomicrons, and lymphocytes [75]. Indeed, this lymphatic dysfunction was demonstrated in a clinical setting by Henriksen, who showed that the lymphatic conductivity in the thoracic duct was significantly lower in patients with cirrhosis and ascites compared to that in healthy individuals [76]. The formation of new lymphatic vessels in adults is rare under normal conditions, while it is very common in various diseases, such as organ rejection, in neoplastic metastasis, and in lymphedema [77,78]. Several molecules are involved in regulating lymphangiogenesis. A common factor in these conditions is the activation of the VEGF pathway, which includes ligands such as VEGF-C and VEGF-D and their receptors (VEGFR2, VEGFR3), intracellular signaling enzymes, such as phosphoinositide 3-kinase (PI3K), and transcription factors, such as Prox-1 [79]. VEGFR3 is the most important receptor in modulating lymphangiogenesis and is highly selectively expressed on endothelial cells committed to the lymphatic lineage. Lymphangiogenesis is driven by VEGF-C and -D, which activate VEGFR3 [80]. This interaction promotes the proliferation, migration, and survival of lymphatic endothelial cells. Prox1 is a transcription factor critical to the development and maintenance of the lymphatic endothelial cells. It regulates the expression of genes involved in lymphatic differentiation and function [81,82]. Podoplanin is a glycoprotein expressed on the surface of the lymphatic endothelial cells. It plays a role in cell adhesion, migration, and signaling during lymphangiogenesis [83]. Angiopoietin-1 (Ang-1) and Angiopoietin-2 (Ang-2) are growth factors involved in the regulation of angiogenesis and lymphangiogenesis. They interact with the Tie-2 receptor on endothelial cells, influencing vessel stability and remodeling [84,85]. Certain fibroblast growth factors (FGFs), such as FGF-2, have been implicated in promoting lymphangiogenesis via VEGF C by stimulating the growth and migration of lymphatic endothelial cells [86]. TGF-β is a multifunctional cytokine that can have both pro- and anti-lymphangiogenic effects depending on the context. It regulates various aspects of lymphangiogenesis, including endothelial cell proliferation and extracellular matrix production [87]. Chemokines, such as CC chemokine ligand 21 (CCL21) and CXC chemokine ligand 12 (CXCL12), play a role in attracting and guiding LECs during lymphangiogenesis [88,89]. PDGF-BB is also implicated in promoting lymphangiogenesis by stimulating the recruitment of pericytes to the developing lymphatic vessels [90]. Although lymphangiogenesis and lymphatic dysfunction seem to have a role in ascites formation, the precise mechanism through which this happens still needs to be elucidated. However, many molecules involved in lymphatic regulation also play a role in ascites development. The main mechanisms mentioned are summarized in Table 5.

## 7. Conclusions

Ascites dramatically impacts the quality of life and prognosis of patients with cirrhosis. Despite extensive research, the complexity of ascites pathophysiology remains incompletely understood. Traditionally, hemodynamic alterations were considered to drive peritoneal fluid accumulation (underfill/overfill theories) [91]. However, recent studies have unveiled a more intricate scenario involving molecular and cellular mechanisms. Inflammation seems to be a key player [2]. Indeed, a complex network of signaling molecules seems to orchestrate the interaction of several cell types involved in this process. Cytokines, chemokines, NO, and growth factors mediate the crosstalk of the involved cells, such as HSCs, SECs, SMCs, liver-resident macrophages, and PMs. The lymphatic system, far from being relegated to a secondary role, displays its cruciality. In cirrhosis, lymphatic vessels lose their normal function, and dysregulated lymphangiogenesis is activated. An increase in vascular permeability, driven by pro-inflammatory cytokines and NO, and impaired lymphatic reabsorption lead to peritoneal fluid accumulation. As a consequence, a systemic hemodynamic imbalance induces plasma volume depletion and a compensatory increase in renal sodium retention. Meanwhile, cytokines can directly activate the RAAS, worsening fluid accumulation. Both gut microbiota and intestinal barrier alterations that occur in cirrhosis are also implicated. Indeed, bacterial and PM translocation in the portal system increase cytokine production. Taken together, all of these factors contribute to ascites formation, each with specific mechanisms which are complicated by their interplay with the others. While some of them have a well-clarified role, others require further investigation. For this purpose, replicating the interaction of these key elements in a controlled setting is warranted, such as in an organ-on-chip model, which could provide more insights into the liver–peritoneal axis.

## Figures and Tables

**Figure 1 biomedicines-13-00680-f001:**
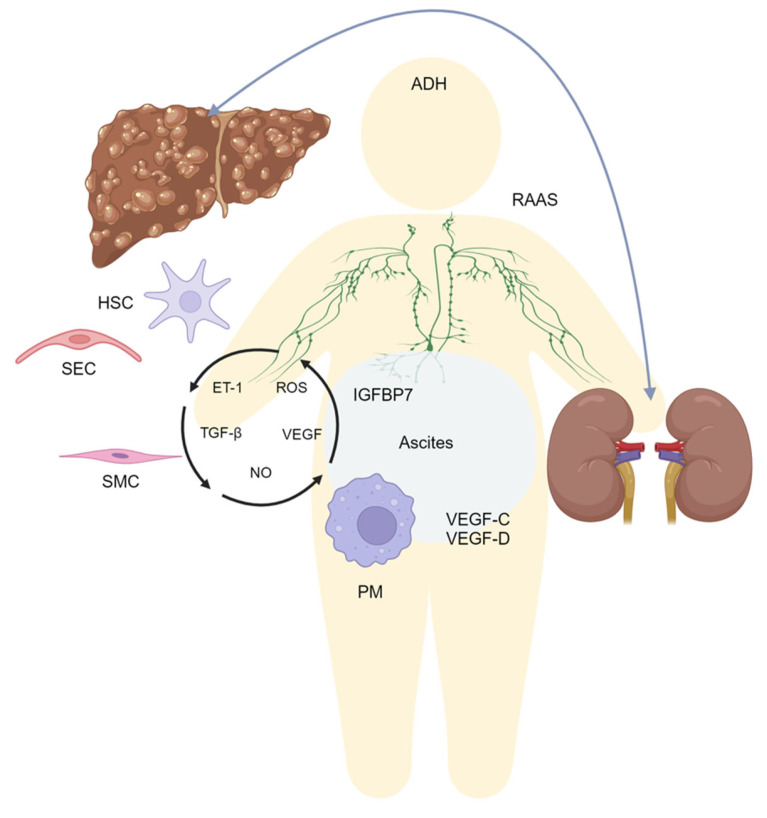
Physiopathology of ascites formation in cirrhotic patients. Ascites develops depending on complex and multifactorial mechanisms that involve an increase in hydrostatic pressure due to portal hypertension, which induces splanchnic vasodilatation, followed by the hyperactivation of the renin–angiotensin–aldosterone system (RAAS) and the increased secretion of antidiuretic hormone (ADH). As a consequence, sodium and water are retained in a vicious circle that increases ascites. Besides this, the alteration in lymphangiogenesis disrupts lymph reabsorption. Furthermore, various hepatic and peritoneal cells are involved, such as hepatic stellate cells (HSCs), sinusoidal endothelial cells (SECs), smooth muscle cells (SMCs), and peritoneal macrophages (PMs), being responsible for intricate crosstalk. Paracrine and endocrine signaling participate in the altered regulation of these actors contributing to ascites formation. IGFBP7: insulin-like growth factor-binding protein 7; ROS: reactive oxygen species; TGF-β: transforming growth factor β; NO: nitric oxide; ET-1: endothelin 1; VEGF: vascular endothelial growth factor. Created using BioRender.com (accessed on 28 August 2024).

**Figure 2 biomedicines-13-00680-f002:**
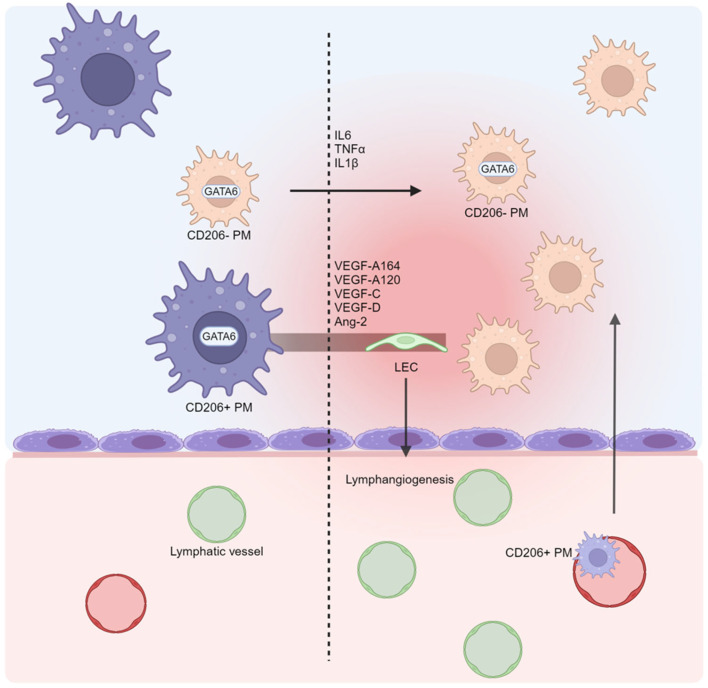
The role of peritoneal macrophages (PMs) in ascites formation. PMs are the main cell type in the peritoneal cavity. In physiological conditions, large CD206^+^ PMs are prevalent, while small CD206^−^ PMs become the predominant cells in response to inflammation. In fact, under inflammation conditions, large PMs migrate to other sites or transdifferentiate into lymphatic endothelial cells (LECs). PM transdifferentiation is sustained by the expression of various pro-lymphangiogenic genes, such as vascular endothelial growth factors (VEGFs) and angiopoietin 2 (Ang-2). This results in an abnormal process of lymphangiogenesis which impairs lymphatic drainage and promotes ascites formation. Besides this, the pro-inflammatory cytokines produced by the PMs, including tumor necrosis factor α (TNF-α), interleukin 1β (IL-1β), and IL-6, increase vascular permeability, renin–angiotensin–aldosterone system (RAAS) activation, and macrophage recruitment. Created using BioRender.com (accessed on 28 August 2024).

**Table 1 biomedicines-13-00680-t001:** Summary of the main mechanisms about nitric oxide (NO) and vascular endothelial growth factor (VEGF).

Study Reference	Study Type and Model	Mechanism Demonstrated
Abraldes et al. (2006) [12]	Preclinical, rat model	Moderate portal hypertension increases VEGF synthesis and eNOS expression, contributing to vascular permeability and ascites formation.
Fernández-Varo et al. (2010) [13]	Gene transfer in cirrhotic rats	Inhibition of eNOS activity reduces ascites volume and improves renal excretion.
Ribera et al. (2013) [14]	In vivo, cirrhotic rat model	NO production in LECs impairs lymphatic drainage, contributing to fluid accumulation.
Masoumi et al. (2015) [15]	Review of clinical and preclinical studies	NO inhibits aldosterone production, promoting natriuresis and potentially alleviating ascites.

Abbreviations. NO: Nitric oxide; VEGF: Vascular endothelial growth factor; eNOS: Endothelial nitric oxide synthase; LECs: Lymphatic endothelial cells.

**Table 2 biomedicines-13-00680-t002:** Summary of the main mechanisms about cytokines and signaling molecules.

Study Reference	Study Type and Model	Mechanism Demonstrated
Albillos et al. (2014) [16]	Clinical study, cirrhotic patients	Cytokines disrupt endothelial barriers, increasing vascular permeability and ascites.
Norlander & Madhur (2017) [17]	Review of molecular pathways	IL-6 and IFN-γ activate the RAAS system, worsening fluid retention.
Kolomeyevskaya et al. (2015) [19]	Clinical study, ovarian cancer patients	Interaction of TNF-α and IL-6 predicts increased vascular permeability and reduced progression-free survival, suggesting a role in ascites formation.
Eriksson et al. (2004) [22]	Clinical study, plasma and ascites analysis	Elevated IL-1α, IL-6, and TNF-α in ascitic fluid correlate with increased vascular permeability.

Abbreviations. IL-1α: Interleukin-1 alpha; IL-6: Interleukin-6; TNF-α: Tumor necrosis factor-alpha; IFN-γ: Interferon γ; RAAS: Renin-angiotensin-aldosterone system.

**Table 3 biomedicines-13-00680-t003:** Summary of the main mechanisms about Hepatic Stellate Cells (HSCs) and Sinusoidal Endothelial Cells (SECs) Crosstalk.

Study Reference	Study Type and Model	Mechanism Demonstrated
Leeming et al. (2013) [30]	Clinical biomarkers study	ECM deposition by activated HSCs correlates with portal hypertension, contributing to ascites.
Du & Wang (2022) [31]	Review of cellular interactions	HSCs release TGF-β and PDGF, enhancing SECs’ ET-1 production, increasing intrahepatic resistance.
Roehlen et al. (2020) [35]	Review of molecular pathways	TGF-β/Smad signaling promotes pro-fibrotic phenotype in SECs, aggravating ascites.
Garbuzenko (2022) [38]	Review of oxidative stress mechanisms	ROS-mediated signaling between HSCs and SECs perpetuates inflammation and fibrosis, worsening fluid accumulation.

Abbreviations. ECM: Extracellular matrix; HSCs: Hepatic stellate cells; TGF-β: Transforming growth factor-beta; PDGF: Platelet-derived growth factor; ET-1: Endothelin-1; SECs: Sinusoidal endothelial cells; ROS: Reactive oxygen species.

**Table 4 biomedicines-13-00680-t004:** Summary of the main mechanisms about macrophages in ascites formation.

Study Reference	Study Type and Model	Mechanism Demonstrated
Kerjaschki (2005) [49]	Review of lymphangiogenesis	PMs transdifferentiate into LECs, enhancing abnormal lymphangiogenesis and impairing drainage.
Kim et al. (2009) [50]	Preclinical, mouse model	Inflammatory stimuli induce PMs’ production of lymphangiogenic VEGFs, worsening ascites.
Tapia-Abellán et al. (2012) [56]	Clinical observation, alcohol-related cirrhosis	Pro-inflammatory cytokine production by PMs in alcohol-related cirrhosis activates ERK1/2 signaling.
Chang et al. (2022) [59]	Preclinical, mouse model	Migration of PMs to the liver influences hepatic macrophage activity, contributing to ascites formation.

Abbreviations. PM: peritoneal macrophage; VEGF: Vascular endothelial growth factor; ERK1/2: Extracellular signal-related kinase 1 and 2.

**Table 5 biomedicines-13-00680-t005:** Summary of the main mechanisms about lymphatic system and lymphangiogenesis.

Study Reference	Study Type and Model	Mechanism Demonstrated
Yamauchi et al. (1998) [74]	Clinical, morphometric analysis	Enlarged hepatic lymphatic vessels in cirrhosis lead to increased lymphatic pressure and ascites.
Henriksen (1985) [76]	Clinical study, thoracic duct conductivity	Reduced lymphatic conductivity in cirrhotic patients with ascites impairs fluid drainage.
Kumar et al. (2021) [75]	Review of lymphatic dysfunction	Dysregulated lymphangiogenesis in cirrhosis impairs lymphatic outflow, contributing to ascites.
Stacker & Achen (2018) [80]	Review of VEGF signaling	VEGF-C and VEGF-D mediated lymphangiogenesis exacerbates fluid accumulation.

Abbreviations. VEGF-C: Vascular endothelial growth factor-C; VEGF-D: Vascular endothelial growth factor-D.

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
