# Peer review of "The Covert Side of Ascites in Cirrhosis: Cellular and Molecular Aspects"

_biomedicines, 2025, doi:10.3390/biomedicines13030680_

Round 1
Reviewer 1 Report
Comments and Suggestions for Authors
An interesting article summarizing relatively less discussed aspects of the pathophysiology of ascites
The manuscript reads well, however, the data about the observations is lacking
The following suggestions are offered:
1. The available data needs to be depicted in tabular form for all the observations mentioned in the review
2. The following may be discussed and referenced :
a. Li Z, Zhu J, Ouyang H. Recent insights into contributing factors in the pathogenesis of cirrhotic ascites. Front Med (Lausanne). 2024 Sep 13;11:1376217. doi: 10.3389/fmed.2024.1376217. PMID: 39346937; PMCID: PMC11427383.
3. The limitations if any should be mentioned
Author Response
Thank you for the comment. It is very constructive and it greatly enriches the quality of our review.
- We provide a table for each paragraph in order to summarize all the reported mechanisms.
- We discussed the cited article, which significantly enriched our argumentation.
Reviewer 2 Report
Comments and Suggestions for Authors
This literature review aims to unravel the molecular mechanisms involved in signaling pathways and cell processes. The authors should address some aspects.
- The title is unsuitable for the literature review since no new pathophysiological model or biochemical mechanism different from those published for decades is discovered or elucidated.
- The abstract should be structured, indicating the review's objective.
- The authors speak of undiscovered aspects of ascitic pathophysiology; perhaps it is hazardous to talk in these terms when many research teams study all the elements in this review.
- The introduction is too long, and the bibliographic references are scarce due to the large amount of information the authors introduce in the text. The introduction should be structured and contextualized to the proposed objective.
- The review is organized into different well-defined theoretical frameworks that would benefit from bibliographic tables and diagrams that summarize and facilitate the reader's current state of the topic. The molecular mechanisms proposed for various cell types would benefit from mechanistic drawings of cell dialogue.
- Many statements made by the authors throughout the text are not referenced, for example, in lines 158, 161, and 202.....
- There are molecular aspects that are out of place to explain in the review since they are fundamental aspects of molecular biology. Line 203-205.
- The conclusions are vague, general, and diffuse.
Author Response
Thank you for the comment. It is very constructive and it greatly enriches the quality of our review.
- We acknowledge the reviewer’s concern and have modified the title to better reflect the nature of our literature review, avoiding any implication of novel pathophysiological models or biochemical mechanisms.
- The abstract has been restructured to clearly indicate the objective of the review, ensuring that the purpose and scope are explicitly stated.
- We have revised the wording regarding undiscovered aspects of ascitic pathophysiology to avoid overstating the novelty of the review while recognizing ongoing research efforts in the field.
- The introduction has been shortened, structured more clearly, and contextualized to align with the review’s objective. Additionally, we have expanded the number of bibliographic references to support the information presented.
- To enhance clarity, we have incorporated bibliographic tables that summarize key concepts.
- Missing citations have been added throughout the manuscript, including in lines 158, 161, and 202, ensuring that all statements are properly supported by references.
- The molecular aspects that were deemed fundamental to molecular biology (lines 203-205) have been revised, with unnecessary explanations removed to maintain focus on the review’s primary objectives.
- The conclusions have been rewritten to be more precise, summarizing the key findings of the review and providing clear take-home messages.
Reviewer 3 Report
Comments and Suggestions for Authors
The article by Airola and collaborators is well written and well argued and is advisable for publication. In order to slightly improve the article, I suggest only two notes
First is that from the 5th page of the manuscript the margin justification to the right is lost and you should correct it.
The second point is that some important references to ascites, liver transplantation and sex influence are missing, such as:
Author Response
Reviewer 3
Thank you for the comment. It is very constructive and it greatly enriches the quality of our review.
- We have corrected the margins’ format.
- We discussed the cited article, which significantly enriched our argumentation.